

# Hydrodynamic and Primary Production Effects on Seasonal DO Variability in the Danube River

Jan Maier[1], Anna-Neva Visser[1], Christina M. Schubert[1], Simon T. Wander[1], Johannes A. C. Barth[1]

[1]Applied Geology, GeoZentrum Nordbayern, Friedrich-Alexander-University Erlangen-Nürnberg, Erlangen 91054, Germany

*Correspondence to*: Jan Maier (jan.m.maier@fau.de)

## Abstract

Dissolved oxygen (DO) is a fundamental indicator for water quality and ecosystem health, particularly in the context of anthropogenic impacts and climate change. This study presents the first large-scale dataset of DO concentration combined with its stable oxygen isotope ratios (expressed as $\delta^{18}O_{DO}$), particulate organic carbon concentrations (POC) and respiration/ photosynthesis (R/P ratios) from five seasonal campaigns along the entire Danube River in 2023 and 2024. Our findings reveal pronounced seasonal DO dynamics driven by temperature, biological activity and hydrodynamic conditions. During spring and summer, enhanced photosynthesis increased DO up to 0.40 mmol/L with $\delta^{18}O_{DO}$ values down to +12.1 ‰ and POC up to 0.25 mmol/L in two highly productive river sections. The strong correlation between $\delta^{18}O_{DO}$ and POC further confirms the dominant influence of primary producers (i.e., photosynthetic organisms) in a river section where a reduced slope led to slower flow and lower turbulence. Notably, $\delta^{18}O_{DO}$ values were significantly lower than those expected for atmospheric equilibrium (+24.6 ‰ ± 0.4 ‰), a pattern rarely documented in large river systems. In contrast, tributary inflows from the Tisa and Sava rivers diluted biomass and organic material inputs and led to declines in DO and POC. By late summer, intensified respiration reversed photosynthetic signals, led to the lowest DO concentrations down to 0.16 mmol/L and raised $\delta^{18}O_{DO}$ up to +23.7 ‰, particularly in the Sava River. In fall, DO levels partially recovered despite continued respiration, while in winter, oxygen input from the atmosphere became the dominant control with minimal biological influences. Overall, this study provides new insights into dynamic interplays between oxygen sources and sinks across the river continuum over seasons. These new insights underscore the need for continuous DO monitoring, particularly in late summer when oxygen levels can become critically low. Understanding these interactions can help to establish efficient aqueous ecosystem management and conservation strategies in the face of environmental and climate change.

## 1. Introduction

Freshwater ecosystems are increasingly threatened by complex global stressors (Borgwardt et al., 2019; Vörösmarty et al., 2013). Beyond climate change, anthropogenically-induced alterations of river morphology (Belletti et al., 2015), disrupted flow regimes (Acreman and Dunbar, 2004; Poff and Zimmerman, 2010), nutrient input (Fowler et al., 2013; Seitzinger et al.,




2006; Sutton et al., 2011), and pollution from urban and industrial sources (Nyenje et al., 2010; Qing et al., 2015; Suthar et al., 2009; Xia et al., 2016) continue to degrade water quality and stability of aquatic habitats. Land use changes further intensify these pressures, thus jeopardizing river basin health. Such influences can also amplify the impacts of environmental disasters (Honisch et al., 2002; Hua, 2017). Consequently, between 2015 and 2021, only 37 % of all European surface water bodies achieved a 'good' ecological status and only 29 % a 'good' chemical status (EEA, 2021) which highlights the urgency to address

these challenges. Particularly excessive agricultural nutrient inputs, primarily phosphorus and nitrogen, threaten aquatic life and can trigger algal blooms that lead to eutrophication, oxygen depletion and ultimately a decline in biodiversity (Carpenter et al., 1998; Dudgeon et al., 2006; Grizzetti et al., 2017).

Among many consequences of these stressors, disruptions of dissolved oxygen (DO) sources and sinks can significantly impact freshwater habitats and influence biodiversity, biogeochemical cycles and overall environmental health

(Franklin, 2014; Killgore and Hoover, 2001; North et al., 2014). Primary producers, such as phytoplankton, serve as a key source of DO via photosynthesis, whereas respiration by heterotrophic organisms acts as major sinks that consume DO (Heddam, 2014; Wetzel, 2011). However, DO concentrations are not solely regulated by biological activity, but also by atmospheric exchange and environmental factors such as temperature, light availability and nutrient levels (Benson et al., 1979; Odum, 1956; Stumm and Morgan, 1995).

In the context of fluvial systems, a better understanding and prediction of DO distributions and their controlling processes are essential to assess aquatic health and to prevent or at least manage potential anoxic events. This is especially important for the Danube River, Europe's second-longest waterway. While the river provides important services for agriculture and energy production, these activities also impose significant pressures on its natural system. Additionally, the Danube serves as a crucial ecological corridor that promotes biodiversity across central and eastern Europe (Habersack et al., 2016; ICPDR,

2015; Sommerwerk et al., 2009). Although the ecological quality improved in the last 30 years, the Danube River still faces ongoing threats, particularly from organic pollution downstream of major cities and key tributaries (Mănoiu and Crăciun, 2021; Wachs, 1997).

Despite the critical role of DO in freshwater ecosystems, research has primarily focused on DO concentration patterns as an indicator of water quality, while the relative contributions of biological and atmospheric inputs remain poorly quantified.

In particular, photosynthesis and community respiration drive DO dynamics in fundamentally different ways, yet their individual effects cannot be fully distinguished through concentration measurements alone. Traditional approaches lack a clear framework to separate DO sources and sinks, thus leaving significant gaps in our understanding. Stable isotope measurements of DO provide a powerful tool to overcome these limitations by distinguishing between three key processes that govern DO dynamics in aqueous environments: photosynthesis, respiration and atmospheric exchange. Aquatic photosynthesis, by

splitting water molecules, transfers a typically [16]O-enriched signature into the DO pool (Guy et al., 1993; Limburg et al., 1999). However, under low DO concentrations, concurrent DO consumption can lead to [18]O-enrichments even during photosynthesis. Similarly, aerobic respiration preferentially consumes [16]O, leaving the remaining DO enriched in [18]O (Mader et al., 2017).



This study is among the few to apply DO isotopes to river systems (e.g., Parker et al., 2010; Quay et al., 1995; Tobias et al., 2007; Wassenaar et al., 2010) and contributes to a still limited body of research.

Given these challenges, a comprehensive understanding of DO sources and sinks in the Danube is essential for assessing and managing river health. To address this, we conducted the first large-scale study of the DO budget of the entire Danube main channel and its key tributaries (e.g., Inn, Thisa, Sava) (Figure 1), based on five sampling campaigns between 2023 and 2024. By integrating high-resolution DO measurements with $\delta^{18}O_{DO}$ analyses, particulate organic carbon (POC), and ratios between respiration and photosynthesis (R/P), we were able to reveal DO sources and sinks along the river continuum

together with its seasonal dynamics. approach could identify critical periods and regions characterized by elevated or depleted DO levels and also disentangled variable contributions by photosynthesis, respiration and atmospheric exchange. These novel findings enhance our understanding of the Danube's functioning and contribute to a strong scientific foundation for river management and conservation strategies amid growing environmental pressures.

## 2. Material and methods

**2.1 Study area**

The Danube has a total length of 2,857 km and a mean annual discharge of 6486 m$^3$/s (Sommerwerk et al., 2009). Its catchment area of 807,827 km² hosts a population of approximately 83 million people in 10 different countries and serves as a vital freshwater resource for Central and Eastern Europe (Habersack et al., 2016; ICPDR, 2015; Sommerwerk et al., 2009). Between July 2023 and September 2024, five sampling campaigns in spring, summer, late summer and fall and winter were

conducted on the main river and the main tributaries (e.g., Inn, Tisa, Sava). During each sampling campaign, between 54 to 89 sampling locations along the entire mainstream were surveyed (Figure 1; dataset will be uploaded on PANGAEA). The coordinates of the sampling sites were recorded in the field with a GPS device, and elevation data were determined with a barometric altimeter (Supplementary Material Table 1). Discharge data were provided by the ICPDR database, https://www.danubehis.org (ICPDR, 2025; last access: 11. March. 2025).

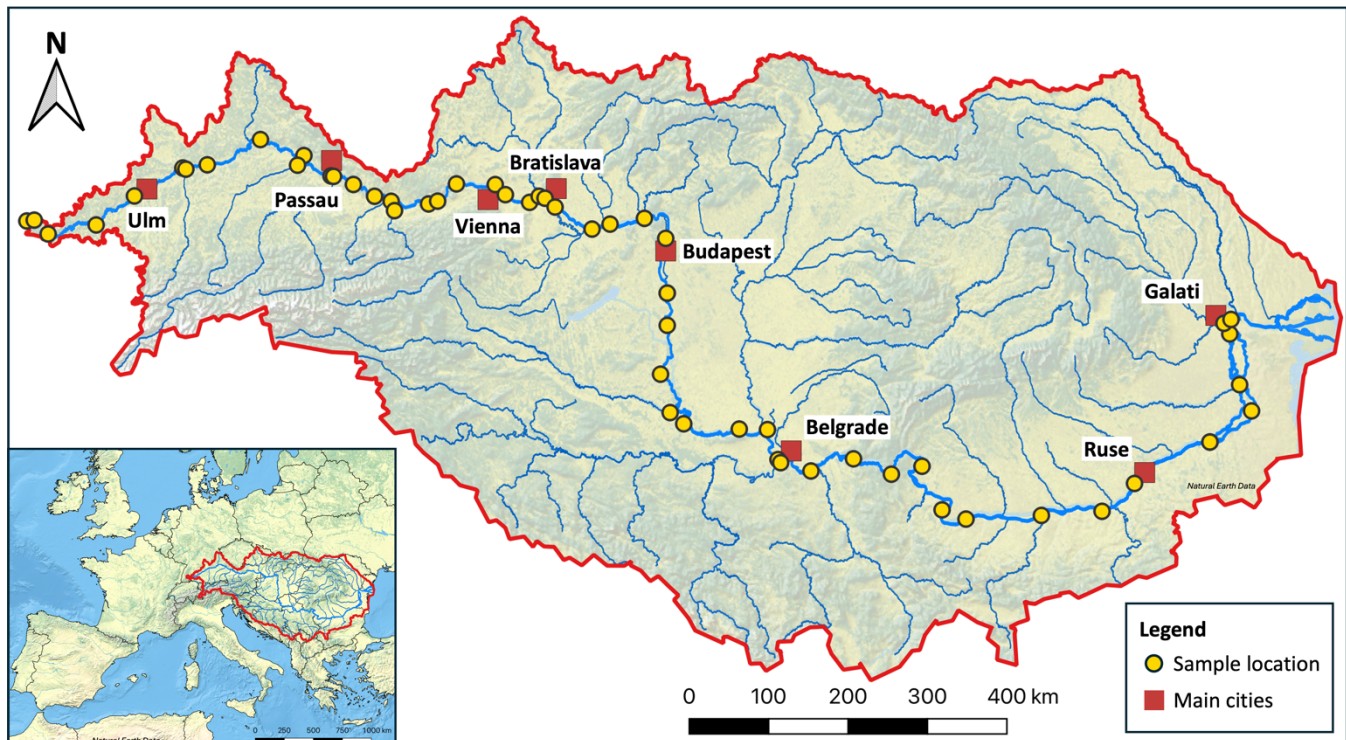

**Figure 1: Map of the Danube River Catchment and its most important tributaries. Danube River Basin District (red), as well as the Danube River and tributaries, data provided by the ICPDR** (last access: 11. March 2025 ICPDR, 2025)**. Yellow dots represent the sampling locations of Late summer 2024 sampling campaign, red squares show main cities along the Danube. The map was created using QGIS v 3.28.3 from © natural earth data (raster data (version 3.2.0 (last access 11 December 2024): https://www.naturalearthdata.com/downloads/10m-raster-data/10m-cross-blend-hypso/) and geoBoundaries (shapefile data (last access 11 December 2024): https://www.geoboundaries.org/globalDownloads.html.**

## 2.2 Field methods

Samples were collected using a weighted 2 L sampling bottle that was submerged between 1 and 2 meters below the water surface to minimize influences of rain and evaporation. Sampling took place either from the center (via bridges or passenger boats) or occasionally from the riverbank. In each case, well-mixed water samples were obtained. To ensure that the samples represented the entire river section, two cross-sectional profiles were taken during each sampling campaign and in all cases, they confirmed homogeneous water mixing.

*In situ* measurements of temperature (T), DO, and oxygen saturation (DO %) were performed with a multiparameter instrument (HQ40d, HACH™, Loveland, CO, USA). This instrument was calibrated daily. Measurement standard deviations were ± 0.1 mg/L for DO, ± 0.42 % for DO %, and ± 0.07 °C for T.

For $\delta^{18}O_{DO}$ analyses, samples were filtered through 0.45 μm pore size nylon syringe filters (Sartorius™) into pre-poisoned 12-mL vials (Labco LTD. Lampeter Exetainer™). These vials contained 10 μL of a saturated $HgCl_2$ solution to



inhibit microbial activity after sampling. Vials were filled completely and sealed with screw caps and butyl septa for efficient

seal-off against atmospheric influences. Triplicate samples were stored in the dark at 4 °C.

For water isotopes (expressed as $\delta^{18}O_{H2O}$), necessary for R/P calculations, water samples were collected in identical

12-mL exetainers without $HgCl_2$ poisoning.

For the determination of POC, 500 mL of unfiltered water samples were collected in acid-washed high-density

polyethylene Nalgene bottles. Before sampling, the bottles were thoroughly rinsed three times with sample water. Preparation

for POC analysis involved filtering the collected water through pre-weighed glass fiber filters (GF-5, pore size 0.4 μm;

Macherey-Nagel, Düren, Germany). To eliminate residual organic carbon, these filters were heated at 400 °C for 4 h and stored

under sterile conditions until sampling.

## 2.3 Laboratory methods

$\delta^{18}O_{DO}$ measurements were performed using a modified automated equilibration system (Gasbench II, ThermoFisher

Scientific™) connected in continuous flow mode to a DELTA V Advantage isotope ratio mass spectrometer (IRMS,

ThermoFisher Scientific™). The analytical approach was based on methods described by Barth et al. (2004) and Wassenaar

and Koehler (1999). Briefly, a 3 mL pure helium headspace was established in the sample vial, and dissolved gases were

extracted by shaking on an orbital shaker at 250 rotations per minute for 30 minutes. The extracted $O_2$ in the headspace was

separated from nitrogen ($N_2$) and other trace gases using a gas chromatography column (CP-Molsieve 5 Å, 25 m length, 0.53

mm outer diameter, 0.05 mm inner diameter; Agilent™, Santa Clara, CA, USA) before introduction into the isotope ratio mass

spectrometer (IRMS) for analysis. Results are reported as averages of triplicate measurements, with an external reproducibility

better than ± 0.2 ‰ (1σ).

Values of $\delta^{18}O_{H2O}$ were determined by infrared spectroscopy (IRIS) with a Picarro™ analyzer (L 1102-i WS-CRDS,

Santa Clara, CA, USA). The analysis was conducted following the protocol outlined by van Geldern and Barth (2012).

All stable isotope values are reported in the standard δ-notation relative to the Vienna Standard Mean Ocean Water

(VSMOW) and are calculated as:

$$\delta = (R_{sample} / R_{reference} - 1) \tag{1}$$

and then multiplied by 1000 to express them in per mille (‰). R represents the molar ratio of the heavy to light isotopes

($^{18}O/^{16}O$ for oxygen; $^2H/^1H$ and $^{18}O/^{16}O$ for $H_2O$) in the sample and the reference (Coplen, 2011). The ratio of VSMOW is

2005.20 ± 0.43 ppm (Baertschi, 1976). For $\delta^{18}O_{H2O}$, external reproducibility was better than ± 0.1 ‰ (1σ).

For POC determination, filters loaded with particulate material were freeze-dried for 60 minutes under vacuum

conditions (<10 mbar) using a freeze-dryer (Lyovac GT 2 GT 2-E, FinnAqua, Gemini BV, Apeldoorn, Netherlands). The dried

filters were then pulverized for 60 s using a ball mill (CryoMill, Retsch, Verder, Vleuten, Netherlands). To ensure complete

removal of potential carbonate residues, the powdered filter material was fumigated with concentrated HCl in a desiccator for



24 h. After fumigation, aliquots of the prepared filter material were carefully weighed into tin capsules (5×9 mm, IVA Analysentechnik GmbH & Co. KG, Meerbusch, Germany). The carbon content of the samples was then determined with an elemental analyzer (Costech ECS 4010, NC Technologies, Bussero, Italy) coupled in helium continuous flow mode to an IRMS (Delta V plus, ThermoFisher, Bremen, Germany).

### 2.4 Isotope calculations

Evaluations of R/P ratios were determined according to Quay et al. (1995) with:

$$\frac{R}{P} = \frac{\left( \frac{18}{16}O_w * \alpha_p - \frac{18}{16}O_g \right)}{\left( \frac{18}{16}O * \alpha_r - \frac{18}{16}O_g \right)} \tag{2}$$

where $\frac{18}{16}O_w$ and $\frac{18}{16}O$ represent the isotope ratios of oxygen in water and in DO, respectively. The photosynthesis fractionation factor ($\alpha_p$) was assumed to be 1.000 ± 0.003 (Russ et al., 2004; Stevens et al., 1975) and the respiration fractionation factor ($\alpha_r$) is commonly assumed with a value of 0.982 for community respiration (Quay et al., 1995).

The parameter $\frac{18}{16}O_g$, described in equation 3, accounts for the isotope ratio of net air-water fluxes and is calculated as:

$$\frac{18}{16}O_g = \frac{\alpha_g * \left( \frac{18}{16}O_a * \alpha_s - \frac{O_2}{O_{2s}} * \frac{18}{16}O \right)}{\left( 1 - \frac{O_2}{O_{2s}} \right)} \tag{3}$$

With $\alpha_g$ being the fractionation factor for gas transfer velocities (0.9972 at 20 °$C$; Knox et al., 1992) and $\alpha_s$ the fractionation factor for oxygen dissolution in water (1.0007 at 28 °$C$) as calculated by Benson and Krause, (1984). The atmospheric oxygen isotopic ratio $\left( \frac{18}{16}O_a \right)$ is known with a value of + 23.9 ‰ (Dordoni et al., 2022). The $\frac{O_2}{O_{2s}}$ ratio represents the concentration of DO in the sample relative to the maximum temperature-dependent equilibrium concentration after Henry's law. Multiplying this ratio by 100 is equal to DO saturation (DO %) as measured in the field.

### 2.5 Statistical analyses

All statistical analyses were conducted in R (v.4.3.2; R Core Team, 2023) using the lm() function to create the linear model and anova() for variance analysis. The coefficient of determination ($R^2$) was reported to evaluate the proportion of variance in $\delta^{18}O_{DO}$ explained by DO concentration (Figure 6a, b) and POC concentration (Figure 6c, d). To assess the effect of DO/POC concentration on $\delta^{18}O_{DO}$, we performed a one-way analysis of variance (ANOVA; Fisher, 1932) based on the following linear

model:

$$\delta^{18}O_{DO} = \beta_0 + \beta_1 DO/POC + \varepsilon \tag{4}$$



where $\beta_0$ is the intercept, $\beta_1$ the regression coefficient, and $\varepsilon$ the error term. ANOVA was used to determine whether DO/POC concentration explains a significant proportion of the variance in $\delta^{18}O_{DO}$. The p-value from the F-test was used to assess statistical significance with a commonly accepted threshold value $\alpha$ of 0.05.

## 3. Results

To investigate seasonal variations in DO, $\delta^{18}O_{DO}$ and POC concentrations, data were analyzed along the Danube River for five different sampling campaigns in spring, summer, late summer, fall and winter. DO concentrations showed clear seasonal fluctuations and variations, ranging from 0.16 mmol/L to 0.4 mmol/L (5.1 and 12.8 mg/L) (Figure 2a-d). The highest mean DO concentrations occurred during winter (0.36 mmol/L $\pm$ 0.01), while the lowest mean occurred in late summer (0.25 mmol/L $\pm$ 0.03). Spatial variability was most pronounced in summer 2023, late summer 2024 and spring 2024 (Figure 2a, d). In contrast,

concentrations during fall 2023 and winter 2024 showed little variation over the entire course of the river and ranged from ca. 0.3 to 0.35 mmol/L (Figure 2b, c). As indicated by the red arrows in Figure 2a, two distinct DO maxima occurred in summer: the first in the mid Danube (~1220 km; 0.35 mmol/L) and the second in the lower Danube (~440 km; 0.40 mmol/L). Similar but less pronounced patterns were also observed in spring and late summer. However, here the downstream maximum did not exceed the upstream one.





**Figure 2: Dissolved oxygen (DO) concentrations (mmol/L) in the Danube River at various distances from the river mouth. Standard error bars are within the symbol size. Triangles denote tributary samples, and dotted lines indicate the river km where tributaries enter. Vertical solid lines represent the Iron Gate I and II dams. The red arrows show DO maxima and blue arrows DO minima.**

Stable isotope values measured for DO ($\delta^{18}O_{DO}$) ranged from +25.9 ‰ to +12.1 ‰ in the entire dataset (Figure 3a-d). Here, the equilibrium with atmospheric oxygen (+24.6 ‰ ± 0.4 ‰) serves as the boundary value between photosynthesis and respiration (Dordoni et al., 2022). Values below this threshold refer to photosynthesis, while values above it refer to respiration. Consequently, samples from summer 2023, late summer 2024, and spring 2024 predominantly showed photosynthesis-driven signals with $\delta^{18}O_{DO}$ values lower than +24.6 ‰ (Figure 3a, d). Conversely, samples from fall 2023 and winter 2024 often fell into the respiration range, even though they were close to the equilibrium (Figure 3b, c). Variability in $\delta^{18}O_{DO}$ was highest during summer (+19.4 ± 3.2 ‰), late summer (+22.4 ± 2.3 ‰) and spring (+21.9 ± 1.4 ‰). On the other hand, fall and winter exhibited lower variabilities with +24.3 ± 1.1 ‰ and +24.7 ± 0.4 ‰. As shown by the red arrows in Figure 3a, two minima in $\delta^{18}O_{DO}$ were observed in summer at the same locations where DO maxima occurred with +14.8 ‰





and +12.1 ‰. Spring and late summer showed similar patterns with two $\delta^{18}O_{DO}$ minima, but the downstream minima were smaller than the upstream.



**Figure 3: Stable isotopes of dissolved oxygen ($\delta^{18}O_{DO}$) in the Danube River at various distances from the river mouth: Standard error bars are within the symbol size. Triangles denote tributary samples, and dotted lines indicate the river km where tributaries enter. Vertical solid lines represent the Iron Gate I and II dams, and the horizontal solid line at +24.6 ‰ is the equilibrium for atmospheric oxygen. Values >+24.6 ‰ indicate respiration and <+24.6 ‰ photosynthesis. The red arrows show $\delta^{18}O_{DO}$ minima.**

POC concentrations ranged from 0.01 to 0.26 mmol/L (0.1 to 3.1 mg/L) (Figure 4a-d). These values largely followed the same trends as DO concentrations and $\delta^{18}O_{DO}$ values with considerable fluctuations in summer 2023, late summer 2024 and spring 2024. In contrast, POC levels in fall 2023 and winter 2024 were less variable and mostly remained below 0.1 mmol/L. As highlighted by the red arrows in Figure 4a, two maxima in POC concentrations were also observed during summer: the first one in the mid Danube (~1210 km; 0.25 mmol/L) and the second in the lower Danube (~500km; 0.24 mmol/L). Samples from late summer showed a similar trend, with a weaker maximum in the mid and lower Danube. Spring samples also showed two peaks in the middle and lower Danube, where the downstream peak was also smaller than the upstream one.





**Figure 4: Particular organic carbon (POC) concentrations (mmol/L) in the Danube River at various distances from the river mouth. Standard error bars are within the symbol size. Triangles denote tributary samples, and dotted lines indicate the river km where tributaries enter. Vertical solid lines represent the Iron Gate I and II dams. The red arrows indicate POC maxima.**

R/P ratios were not plotted over the entire range of the river but showed similar trends as shown for DO, $\delta^{18}O_{DO}$ and POC (Figure S1). However, a cross plot between $\delta^{18}O_{DO}$ and R/P ratio showed a non-linear relationship of both parameters (Figure 5). R/P ratios ranged from 0.1 in summer to 8.9 in fall. Spring and summer samples were more dominated by photosynthesis, with R/P ratios smaller than 1, while winter and fall samples were more dominated by respiration, with R/P



ratios larger than 1. Late summer samples exhibited a mixed signal, with most values in a range of overlapping photosynthesis

and respiration and R/P values mostly larger than 1.

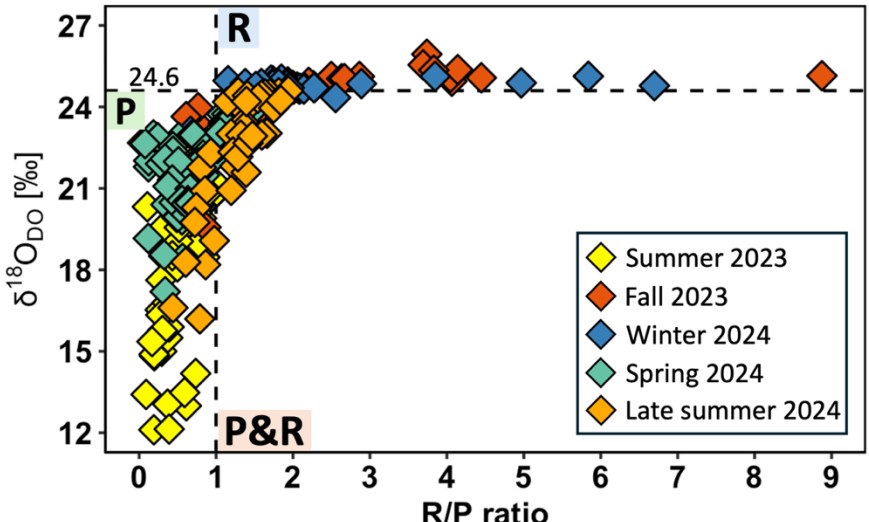

**Figure 5: Cross plots $\delta^{18}O_{DO}$ vs. respiration/photosynthesis (R/P) ratios of the Danube River sampling locations. The horizontal dotted line at +24.6 ‰ marks the equilibrium for atmospheric oxygen. Values with >+24.6 ‰ indicate respiration and <+24.6 ‰ photosynthesis. The vertical dotted line at R/P = 1 denotes the transition between photosynthesis and respiration. R/P values <1 indicate photosynthesis, and R/P >1 respiration.**

210        Correlation plots between $\delta^{18}O_{DO}$ and DO concentrations were created for spring and summer (Figure 6a), and for late summer, fall and winter (Figure 6b). Here strongest correlations were observed in spring ($R^2 = 0.52$, p <0.05) and summer ($R^2 = 0.75$, p <0.05) which were both statistically significant. In spring, higher DO coincided with lower $\delta^{18}O_{DO}$, while in summer, this relationship was even stronger. In contrast, correlations weakened in late summer ($R^2 = 0.14$, p <0.05) and fall ($R^2 = 0.33$, p <0.05), though still statistically significant. This suggests that while $\delta^{18}O_{DO}$ and DO are still related, other factors likely

contribute to $\delta^{18}O_{DO}$ variability. In winter ($R^2 = 0.04$, p >0.05), no significant correlation was observed between both parameters.

       To further investigate potential biological influences on $\delta^{18}O_{DO}$, it was correlated with POC concentrations at selected areas of high DO contents (ca. 1600 to 200 km) that were found during spring and summer (Figure 6c) and for late summer, fall, and winter (Figure 6d). The strongest correlations were observed in spring ($R^2 = 0.60$, p <0.05) and summer ($R^2 = 0.88$, p

<0.05) which were both statistically significant. In spring, this correlation was moderately strong, and increasing POC concentrations coincided with decreasing $\delta^{18}O_{DO}$ values. However, no direct biological influence on $\delta^{18}O_{DO}$ values could be identified in late summer ($R^2 = 0.16$), fall ($R^2 = 0.06$), and winter ($R^2 = 0.13$), as reflected by weak, non-significant correlations (p >0.05). These results suggest that POC does not exert a clear influence on $\delta^{18}O_{DO}$ during these three seasons, although other environmental or biological factors may contribute.



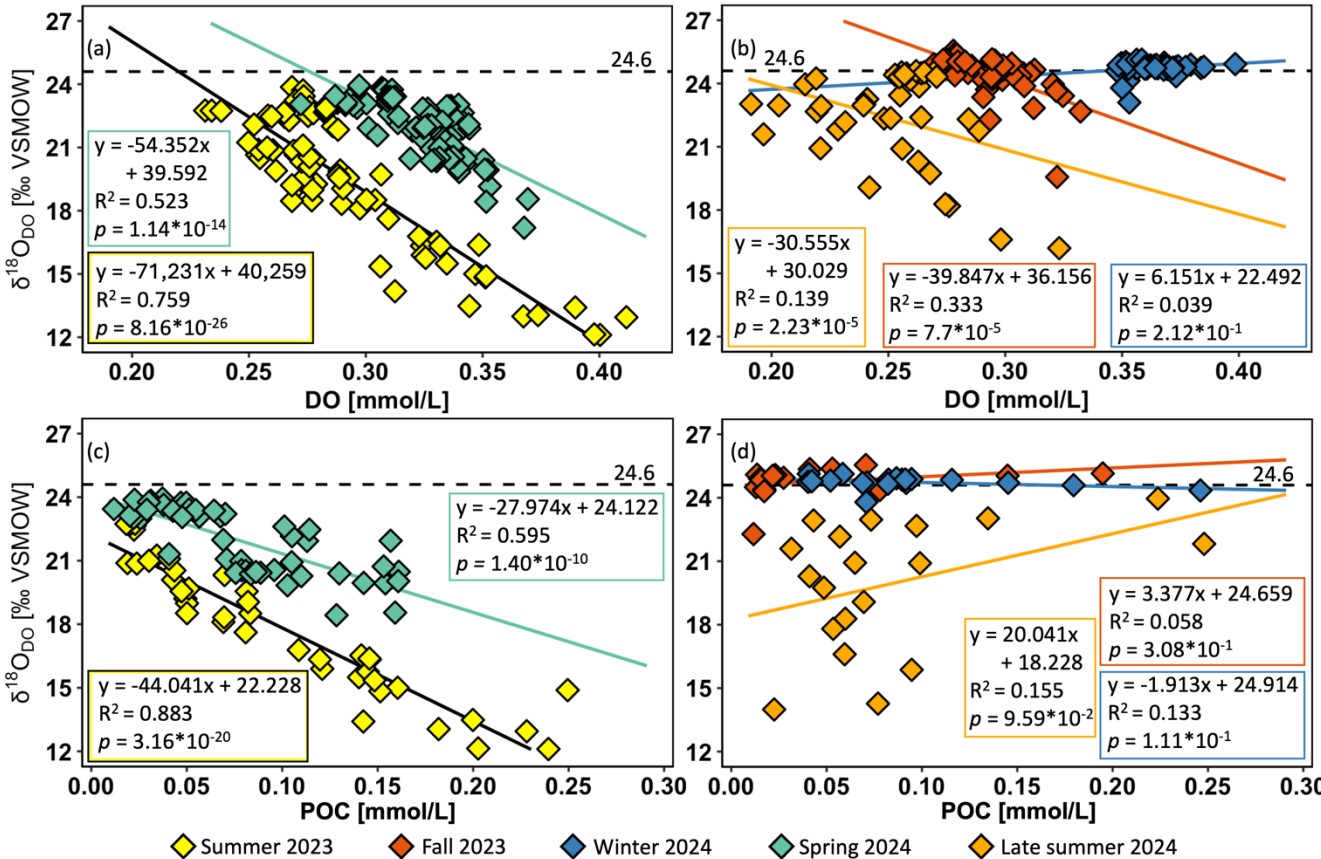

**Figure 6: Correlation plots for all Danube sampling locations of $\delta^{18}O_{DO}$ and dissolved oxygen (DO) in summer and spring (a), in late summer, fall, and winter (b). High productivity areas (~1600 to ~200 km) related correlation plots of $\delta^{18}O_{DO}$ and particulate organic carbon (POC) for the same periods are presented to highlight potential biological inputs (c, d). The negative correlation indicates that as POC increases as $\delta^{18}O_{DO}$ values decrease. This pattern is consistent with biological oxygen production (<+24.6 ‰) and consumption (>+24.6 ‰), as oxygen produced by algal activity is isotopically lighter than that of the atmosphere. These results support the role of algae-derived organic matter in influencing the $\delta^{18}O$ signature of dissolved oxygen.**

## 4. Discussion

### 4.1 Seasonal Dynamics

Seasonal variations in DO and $\delta^{18}O_{DO}$ along the Danube River are primarily influenced by temperature, biological activity, and atmospheric exchange. During winter, colder water temperatures decreased to 6.5 °C and thereby enhancing $O_2$ solubility (Figure S2c) (Rettich et al., 2000). Additionally, higher river discharge and associated turbulence likely further increased $O_2$ solubility and led to the highest DO concentrations measured throughout the year (Figure 2c; Figure S3, S4c) (Vautier et al., 2020). The associated $\delta^{18}O_{DO}$ data support this interpretation of an atmospheric oxygen source as values remained consistently close to those expected for atmospheric equilibrium (±24.6 ‰ ± 0.4 ‰) (Figure 3c; Figure S5). Although respiration must have occurred as well, its overall impact on the DO budget was likely minimal and largely masked by continuous atmospheric





equilibration (Figure 3c, 6b). At first glance, associated R/P ratios might contradict this interpretation with values larger than 1 (Figure 5, Figure S1c). However, according to Dordoni et al. (2022), true dominance of respiration only becomes evident when R/P ratios exceed 2. Therefore, winter conditions reflect a mixed signal, with minor respiration influences and dominant atmospheric exchange.

        With rising temperatures and longer daylight in spring, biological activity became more prominent and led to
moderate to high DO concentrations (Figure 2d; Figure S2d, S3) (Wetzel, 2011). The associated $\delta^{18}O_{DO}$ values indicated enhanced photosynthetic activity, as they gradually shifted from atmospheric equilibrium toward less positive values (Figure 3d; Figure S5). Such a photosynthesis increase was further confirmed by slight rises in POC concentrations when compared to winter. While some of this POC may have originated from external sources, such as soil material washed into the river (Aramaki et al., 2010; Reddy et al., 2021), the concurrent buildup of POC, DO, and $\delta^{18}O_{DO}$ suggests an accumulation of algae
biomass driven by in-river photosynthetic activity (Figures 2d, 3d, 4d). Direct influences of photosynthesis on DO dynamics were further supported by a moderate but statistically significant correlation between DO concentration and their $\delta^{18}O_{DO}$ values ($R^2 = 0.52$, $p < 0.05$), thus highlighting the link between oxygen production and its photosynthetic source. Although photosynthesis predominantly influenced DO in spring, atmospheric exchange likely remained a significant contributor. This process was likely supported by cool temperatures that remained below 18 °C and elevated discharge rates that increased
turbulent flow (Figures S2d, S3d). Moreover, the interplay between biological and physical processes was also reflected by R/P ratios (Figure 5, Figure S1d). Most of these ratios remained below 1 and indicated shifts towards photosynthesis. However, occasional mixed signals, with R/P ratios around and larger than 1, suggest that both atmospheric exchange and respiration processes also influenced the river's DO budget.

        The intensification of biological activity in summer was most evident by $\delta^{18}O_{DO}$ values that decreased to +12.1 ‰
(Figure 3a; Figure S5). This distinct shift marks the time of peak photosynthetic activity in the Danube and is particularly notable because such pronounced deviations from atmospheric equilibrium are typically observed in standing waters (e.g., Quay et al., 1995; Wassenaar, 2012). However, similar deviations caused by photosynthesis have also been recorded in smaller streams (e.g., Parker et al., 2010; Wassenaar et al., 2010)and support our finding that even running waters are capable of strong photosynthetic DO input. The strong photosynthetic signal was further reflected in the increase of POC concentrations,
indicating an accumulation of organic matter. Overall, the simultaneous increases of POC and DO, along with parallel decreases in $\delta^{18}O_{DO}$ and characteristic fluctuations in R/P ratios that also consistently fell below 1 in summer, clearly underscore intensified photosynthetic activity (Figures 2a, 3a, 4a, 5; Figure S1a). The strong linear correlation and statistical significance between $\delta^{18}O_{DO}$ and DO further confirm this relationship ($R^2 = 0.76$, $p < 0.05$; Figure 6a). Interestingly, despite such strong biological DO inputs, DO concentrations were often lower in summer when compared to winter and spring (Figure
S3). These declines in DO concentrations can be attributed to higher water temperatures (Figure S2), which reduce $O_2$ solubility (Rettich et al., 2000). Another factor that may have reduced DO could have been the simultaneous decomposition of POC by respiration. While this process likely contributed to DO depletion, its impact on POC levels was minimal. In fact, POC input appeared to outperform its consumption, as indicated by rising concentrations (Figure 4a). Although DO levels were higher in



winter, photosynthetic DO production likely played an important ecological role during warmer periods when atmospheric $O_2$

contributions were less effective.

Late summer samples differed from those of mid-summer, despite similar water temperatures (Figure S2a). During this period, DO concentrations reached their lowest levels, particularly in the Sava River, where oxygen levels decreased to 0.16 mmol/L (see blue arrow in Figure 2a). After its confluence with the Danube, DO levels showed a slight recovery to around 0.2 mmol/L. Given that healthy aquatic systems should maintain DO levels above 0.156 mmol/L (WHO, Atlas Scientific),

these late summer values raise concerns about critical oxygen depletion. This concern is further supported by a shift in R/P ratios from a photosynthesis-dominated signal (R/P<1) to a mixed signal (R/P>1) (Figure 5; Figure S1a), thus indicating that respiration rates had begun to exceed those of photosynthesis. Additionally, compared to mid-summer, the weakening correlation between DO and $\delta^{18}O_{DO}$ ($R^2 = 0.14$) suggests a reduced influence by photosynthesis (Figure 6b). A combination of declining primary production, enhanced respiration, and potential organic matter decomposition likely contributed to the

observed decrease in DO concentrations. With rising temperatures due to climate change, further DO declines can be expected, particularly during periods of reduced photosynthetic activity, such as late summer. This trend may also apply to other temperate large river systems, when late summer and early fall become the most critical periods for DO depletion (Piatka et al., 2021; Zhi et al., 2023). Although water quality in the Danube has improved over the last three decades, these findings highlight the need for continuous oxygen monitoring, especially in the Sava and Lower Danube regions (Mănoiu and Crăciun,

290 2021).

In fall, declining water temperatures increased $O_2$ solubility and led to a moderate rise in DO concentrations as compared to late summer (Figure 2b; Figure S2b) (Rettich et al., 2000). These temperature decreases also contributed to a more homogeneous DO distribution along the entire river system, which showed a large-scale seasonal effect that contrasted with the localized areas of high DO production in summer. As a result, $\delta^{18}O_{DO}$ values shifted towards more positive values and

approached those of atmospheric exchange (Figure 3b; Figure S5). Additionally, reduced light intensity and a decline in photosynthetic activity during fall also affected the entire river system (Aruga, 1965; Collins and Boylen, 1982). With the decline in photosynthesis, respiration became the dominant process, as also reflected by R/P ratios that were mostly above 2, especially in the lower section of the Danube (Figure 5; Figure S1b). This transition from a photosynthesis to a respiration-dominated system can be attributed to the accumulation of organic material over the summer, which provided fresh biomass

for decomposition through respiration in fall. (DeNicola, 1996; Uehlinger et al., 2000). Although in fall increased respiration was evident in the R/P ratios, $\delta^{18}O_{DO}$ levels values themselves remained close to atmospheric equilibrium (Figure 3b; Figure S5). This pattern suggests that the Danube system is buffered by atmospheric $O_2$ exchange and serves as a crucial DO source that prevents DO depletion despite the dominance of respiration.

Overall, seasonal fluctuations in DO and $\delta^{18}O_{DO}$ along the Danube River were primarily influenced by temperature,

associated atmospheric exchange and biological activity. In spring and summer, photosynthesis was an important process that increased DO and lowered $\delta^{18}O_{DO}$, while in late summer, enhanced respiration became predominant and caused lower DO levels and higher $\delta^{18}O_{DO}$, particularly evident in the Sava tributary. With cooler temperatures in fall, DO levels recovered,



although respiration remained an important driver. In winter, oxygen solubility was the dominant control, with minimal biological influence.

## 4.2 Areas of increased primary production

In addition to the seasonal dynamics observed throughout the year, two distinct areas of increased DO production emerged during the warm season. One in the middle and the second in the lower section of the river. These areas exhibited pronounced increases in all three parameters: DO, $\delta^{18}O_{DO}$ and POC concentrations (Figures 2a, d; 3a, d; 4a, d; Figure S6a-c). Here, the pronounced shifts in $\delta^{18}O_{DO}$ values down to +12.1 ‰ are most remarkable, because such large deviations from atmospheric equilibrium are typically observed in standing waters (Quay et al., 1995; Wassenaar, 2012) or smaller streams (Parker et al., 2010; Wassenaar et al., 2010), but rarely in larger river systems (Quay et al., 1995). These shifts clearly indicate enhanced photosynthesis, which has already been identified as the primary driver of oxygen production during the warmer months (Figure 6a, b). The strong photosynthetic impact is further supported by clear correlations between $\delta^{18}O_{DO}$ values and POC concentrations (Figure 6c). This relationship indicates the dominance of autotrophic organisms as key DO sources in these highly productive river sections during spring and summer and is supported by previous studies on the Danube, which also emphasized the importance of autotrophic activity (Hein et al., 1999; Riedler P. and Schagerl M., 1998).

As the warm season progressed, the correlation between $\delta^{18}O_{DO}$ and POC strengthened, thus reflecting the increasing role of autotrophic production, with a moderate correlation in spring ($R^2 = 0.60$) that became even stronger in summer ($R^2 = 0.88$). These positive correlations indicate that a substantial portion of the organic material must have originated from primary producers and directly contributed to the formation of these maxima. Although the same areas of elevated productivity did not change in late summer (Figure 2a, 3a, 4a), no significant correlation could be observed during this period (Figure 6d). This lack of correlation may result from the stabilization of high DO values during this transition period or the increasing influence of external POC sources, such as material transported into the river. Despite the continued elevated DO levels, this seasonal shift likely reflects a decline in primary production as environmental conditions changed and led to reduced contributions of primary producers to the overall organic material.

The link between primary producers and DO dynamics observed in this study is further supported by previous research on the Danube. For instance, phytoplankton data from Literáthy et al. (2002) showed that the highest biomass occured in the middle section of the river, thus aligning with the area identified in our study and indicating the connection between increased primary production and DO-rich zones. Furthermore, Dokulil, (2015) highlighted findings from several chlorophyll-α studies in the Danube that indicated peak algae growth during summer with significant temporal variations in both the intensity and timing of algae blooms across years (e.g., 1988, 1998 and 2001). The agreement between these findings and our data suggests that algal blooms occur regularly in this section of the Danube, contributing to and reinforcing seasonal variations.

One key factor that influences these areas of increased productivity is the changing flow gradient of the Danube (Figure S6d). According to Habersack et al. (2016), the riverbed slope decreases in its middle section from a steeper gradient of about 0.4 % in the upper Danube to a much flatter gradient of about 0.1 %. This reduction in slope decreases the velocity



of the river and creates flow conditions that favor primary producers. The resulting slower water flow enhances the sedimentation which in turn decreases turbidity and light penetration and further supports photosynthetic activity. These conditions also align with findings of Dokulil (2006, 2015), who identified the middle reach of the Danube as an optimal zone for primary production due to moderate flow velocities and increased light availability. Moreover, the Danube exhibited only

moderate discharge levels during summer that could further intensify these effects (Figure S4a). While not at its lowest, this discharge still enables longer water residence times that increase light exposure for algae and create favorable growth conditions (Kamjunke et al., 2021; Weitere and Arndt, 2002).

     Another factor that may have contributed to the observed productivity patterns is the proximity of Budapest (Figure S6a-d). Located upstream of the productivity maximum in the mid-Danube, it could influence primary production through

nutrient inputs. Urban areas such as Budapest are known sources of nutrient pollution due to wastewater discharge (Nyenje et al., 2010; Xia et al., 2016), while agricultural activities further contribute to nutrient loading. However, our analyses did not reveal clear links between nutrient concentrations, POC, DO levels or $\delta^{18}O_{DO}$ values. For instance, nitrate concentrations ranged around a mean value of 0.1 mmol/L ± 0.0025 (standard error) throughout the year and along the entire river (Figure S7a-d), while phosphate levels remained consistently below the detection limit. These results align well with previous findings

published by Liška et al. (2021), who reported almost stable nitrate concentrations in the Danube over the past decades without noticeable peaks from potential point sources. One possible explanation is that primary producers rapidly absorb and utilize available nutrients and prevent their accumulation in measurable concentrations (Joint et al., 2001; Wetzel, 2011). This mechanism could explain the observed productivity patterns, despite minimal variance in nutrient levels and may particularly account for phosphate that is rapidly taken up by primary producers. However, to further investigate this possibility, more

detailed studies on aquatic biomass and nutrient uptake processes would be required.

     Following the upstream productivity maximum near Budapest, DO concentrations declined abruptly, together with decreases in POC and increases in $\delta^{18}O_{DO}$ values (Figures 2a, d; 3a, d; 4a, d). Previous studies have also documented reductions in phytoplankton biomass (Dokulil and Kaiblinger, 2008; Literáthy et al., 2002) and chlorophyll-α concentrations (Dokulil, 2006, 2015) in this region. These patterns suggest that external factors, particularly the inflows of the Tisa and Sava rivers,

play a role. Both tributaries introduce large volumes of water and contribute substantial dilution to the main course of the Danube. Although Tisa and Sava rivers could not be sampled during all campaigns, data from late-summer and spring indicate that they contain lower DO concentrations (blue arrow Figure 2a, d), higher $\delta^{18}O_{DO}$ values, and reduced POC levels compared to the main stem of the Danube (Figure 3a, d; 4a, d). This supports dilution as a plausible mechanism for the observed decrease. Further evidence arises from previous studies, which estimated that this confluence with the Danube is being diluted by

approximately 27 % during average discharge conditions (Dokulil, 2015). Discharge data from summer and spring confirm this effect, showing a substantial increase in total river discharge following the confluence with these tributaries (Figure S4a, d). Additionally, the inflow of the Morava River (not analyzed in this study) likely added further dilution effects (Dokulil, 2015).





Further downstream, the Danube featured a second and even more intense productivity zone with similar patterns to
the upstream peak area. While this area shows increased chlorophyll-α and suspended solids (Dokulil, 2015; Dokulil and
Kaiblinger, 2008), phytoplankton biomass does not rise accordingly (Literáthy et al., 2002), suggesting an unconventional
productivity pattern. On the other hand, a pronounced decrease in $\delta^{18}O_{DO}$ indicated active photosynthesis despite potential
phytoplankton growth limitations (Wetzel, 2011). Notably, this region exhibited the lowest $\delta^{18}O_{DO}$ values in the study, along
with R/P ratios close to 0 in summer, thus underscoring the significance of photosynthetic activity (Figure 5; Figure S1a).
These findings align with those of Dokulil (2006, 2015), who reported that despite significant increases in chlorophyll-α,
primary production remained low due to poor light availability in the water column caused by elevated turbidity. Similar to
the upstream peak, the emergence of this second productivity zone coincided with a flattening river gradient (0.05 – 0.01 %)
that likely fosters favorable hydrodynamic conditions for algae (Habersack et al., 2016).

The spatial patterns observed also coincide with the above-discussed seasonality. They enhance spatial DO dynamics
and play a crucial role, as the second most productive area is clearly visible in summer but weakens or even disappears by late
summer (Figure 2a, 3a). Initially, the peak was driven by primary production, but as the season progressed, it declined rapidly.
This decrease was also reflected by the absence of a pronounced $\delta^{18}O_{DO}$ peak and a shift in R/P ratios toward respiration in
late summer (Figure 5b; Figure S1a) highlights growing influences of heterotrophic processes, particularly in the lower section
of the river. Additionally, the decreasing correlation between $\delta^{18}O_{DO}$ and POC concentration in late summer suggests a shift
in dominant respiration processes (Figure 6d). While such a strong correlation in summer indicated active primary production,
this link weakened as phytoplankton started to decompose, thus leading to organic matter accumulation and increased
microbial respiration. This transition further supports the idea that the second peak is initially fueled by photosynthesis during
slow flow but shifts toward organic matter degradation later in the season. Therefore, areas with shallow gradients not only
promote algal growth but also facilitate biomass accumulation and subsequent respiration, making them key zones of
biogeochemical activity in the river.

This study identified two areas of high productivity in the Danube, as marked by elevated DO concentrations, $\delta^{18}O_{DO}$
stable isotopes and POC levels. These maxima occurred in regions with reduced slopes, where slower flow velocities and
lower turbulence likely promoted increased autotrophic activity. In these sections of the river, primary producers emerged as
key drivers of DO input, with a strong correlation between $\delta^{18}O_{DO}$ and POC concentrations in spring and summer. Notably,
$\delta^{18}O_{DO}$ values reached as low as +12.1 ‰, a deviation from atmospheric equilibrium rarely observed in larger river systems.
Connecting these findings with previous research underscores the critical role of primary producers in shaping oxygen
dynamics and influencing organic material distribution across the Danube River. However, the persistence of high algal growth
highlights the need for stricter compliance with EU regulations on water quality. Since DO serves as a key parameter for
assessing ecosystem health, further monitoring and regulation are essential to mitigate human-induced impacts and maintain
balanced aquatic conditions in the Danube system.



## 5. Conclusions

Our study demonstrated that seasonal variations in temperature, biological activity and hydrodynamic conditions drive complex DO dynamics in the Danube River. In spring and summer, enhanced photosynthesis raised DO and POC levels, while lowering $\delta^{18}O_{DO}$ values. This precise source identification is novel in the Danube River and serves as a crucial indicator of ecosystem health. The photosynthetic effects were particularly evident in two areas of higher productivity where reduced slopes resulted in slower flow velocities and lower turbulence. In these zones, where river conditions approached those of standing waters, primary producers played a critical role, as shown by strong correlations between $\delta^{18}O_{DO}$ and POC. These findings underscore the importance of autotrophic DO production in the Danube and suggest that with rising temperatures and resulting reduced DO solubility the ecosystem may increasingly rely on this internal form of oxygen input. However, tributary inflows from the Tisa and Sava rivers diluted biomass and organic material inputs and contributed to declines in DO and POC levels downstream.

In terms of seasonality, late summer emerged as the most critical season of the Danube River, especially the lower part of the Danube River. In contrast, colder seasons of fall and winter facilitated DO recovery and emphasized atmospheric O2 input as another key and temperature-dependent source. With rising global temperatures and warmer cold seasons, future DO budgets could be at risk during these periods. By integrating high-resolution measurements of DO, $\delta^{18}O_{DO}$, POC and R/P ratios, our study identified new arrays of biogeochemical processes that regulate DO dynamics.

Notably, DO concentrations alone are often insufficient to identify their sources, but when combined with $\delta^{18}O_{DO}$ values, relative contributions of different sources (photosynthetic and atmospheric input) and sinks (respiration) become much clearer. Such knowledge is crucial for future river management and may help to plan ecological conservation amid increasing environmental pressures. Thus, we recommend high-resolution monitoring of DO and its sources and sinks, with a particular focus on vulnerable regions with critically low DO in late summer. The potential to identify DO sources and sinks in other aquatic systems with the use of stable isotopes offers a powerful tool for understanding and conserving diverse aquatic ecosystems globally.

## Data availability

We already submitted our data sets from this manuscript to PANGAEA in March 2025. Until its publication all data are available upon request.

## Author contributions

JM: sampling, conceptualization, formal analysis, investigation, methodology, visualization, and writing (original draft preparation). ANV: conceptualization, formal analysis, investigation, methodology, visualization, and writing (original draft preparation). CMS: investigation, methodology, visualization, and writing (original draft preparation). STW: investigation,



methodology, visualization, and writing (original draft preparation). JACB: conceptualization, resources, supervision, and writing (original draft preparation).

**Competing interests**

The contact author has declared that none of the authors has any competing interests.

**Acknowledgements**

This research has been funded by the DALIA project (Danube Region Water Lighthouse Action) – project no. 101094070 that supported work by Friedrich-Alexander-University Erlangen-Nürnberg. Discharge visualization used in this manuscript was produced by the International Commission for the Protection of the Danube River (ICPDR). We acknowledge and appreciate their contribution to the availability and accessibility of valuable data for our research. We thank OpenAI's ChatGPT for
language suggestions and improvements in the writing process. We also express our gratitude to Christian Hanke, Irene Wein, Anja Schuster and Robert van Geldern for providing technical and analytical support. We further thank SE-Tours GmbH and nicko Cruises Schiffreisen GmbH for allowing us to join their cruises at partially reduced prices to conduct our sampling from their ships. Their cooperation has greatly facilitated our research.

**Financial support**

This research has been supported by the DALIA project – project no. 101094070.



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
