# Peer review of "Hydrodynamic and Primary Production Effects on Seasonal DO Variability in the Danube River"

_EGUsphere, 2025_

## Author Response (AR1)

Comments Referee #1:

Review of "**Hydrodynamic and Primary Production Effects on Seasonal DO Variability in the Danube River**"

By Jan Maier, Anna-Neva Visser, Christina M. Schubert, Simon T. Wander, Johannes A. C. Barth

- General comments on manuscript quality and scientific relevance

The manuscript "Hydrodynamic and Primary Production Effects on Seasonal DO Variability in the Danube River" by Maier et al. presents an original case study that meets the criteria of good scientific quality and offers a valuable contribution to our understanding of river systems. It is well structured, and findings are appropriately discussed with reference to relevant literature. The results are clearly outlined and reproducible.

The manuscript focuses on dissolved oxygen (DO) dynamics in the Danube River and offers insights into DO relationship with particulate organic carbon (POC) and the role of respiration/photosynthesis (R/P) ratios. These aspects are of significant relevance to the field of biogeosciences and this contribution is therefore suitable for publication in *Biogeosciences*.

However, there are specific aspects that the authors may wish to elaborate on further (see "Specific comments").

- Specific comments

(19-20) I would report results from fall 2023, as they showed the highest $\delta^{18}O_{DO}$.

**Response:** We would like to thank Marlene Dordoni for her positive feedback and her thoughtful comments. We agree with the reviewer's comment and thus added $\delta^{18}O_{DO}$ values from fall season.

(42) Here it is worth mentioning that chemical processes may become significant DO-sinks too (10.1016/j.jhydrol.2017.01.014 , 10.3133/ofr20091004 , 10.5194/egusphere-egu25-2083)

**Response:** Indeed, it is worth mentioning these processes and we thank the reviewer for her literature suggestions. We changed our text accordingly and implemented the suggested literature.

(75) I am not familiar with this river system and would perhaps have appreciated a more in-depth description of the Danube configuration (e.g. the Iron Gates).

**Response:** We appreciate the reviewer's suggestion and have provided detailed clarification in this response. However, we decided not to include this detail in the manuscript to maintain focus and readability.

As mentioned in Section 2.1 the Danube River is with 2,857 km one of Europe's major waterways and has undergone significant anthropogenic modifications, particularly in regions

where it is partially utilized for hydroelectric power, flood control and other purposes. Consequently, the morphology of the river, including its depth, has undergone significant changes along different sections of its course. The depth of the Danube fluctuates considerably, shaped by factors such as water discharge, sediment transport, and regional geography, with average depths range from approx. 1 to 8 meters. There are, however, notable exceptions, for instance, dammed sections such as the Iron Gate Reservoir can reach depths of up to 120 meters. Historical water levels and river discharge at several monitoring stations along the Danube and its major tributaries are available at https://www.danubehis.org.

Here are a couple of examples for January-July 2024:

- **Achleiten** (Upper Danube, river km 2223): Maximum depth approx. 7.0 m; minimum depth approx. 2.8 m.
- **Mohács (**Middle Danube, river km 1447): Maximum depth approx. 8.0 m; minimum depth approx. 2.8 m.
- **Reni** (Lower Danube, river km 132): Maximum depth approx. 4.0 m; minimum depth approx. 1.1 m.

(93) I would add information regarding the apparatus of the "sampling bottle" used for sampling, and how "well-mixed water samples" were obtained.

**Response:** We thank the reviewer for his suggestion and added the required information.

(223-224) I would move this to the "Discussion".

**Response:** We agree and have moved this content from the Results to the Discussion.

(273) I do agree that this significant DO depletion is linked both to reduced $O_2$ solubility in water and to enhanced organic matter decomposition due to higher temperatures. High levels of POC do not necessarily contradict this interpretation, as sampling frequency may not have allowed a complete record of POC dynamics. Additionally, note that $\delta^{18}O_{DO}$ was proven to be more sensitive to metabolic changes than other chemical parameters (10.1111/jpy.13455).

**Response:** Indeed, the elevated POC levels do not necessarily contradict intensified decomposition activity, particularly in light of limited sampling frequency. We have clarified this point in the Discussion. In addition, we have incorporated the observation that $\delta^{18}O_{DO}$ is more sensitive to metabolic shifts than other chemical parameters, thus allowing for earlier detection of transitions between photosynthesis and respiration.

(324) It might be helpful to briefly clarify which types of primary producers are being referred to. When multiple phytoplankton communities coexist, different groups can exhibit diverse photosynthetic rates. This can lead to varying contributions to the DO pool. This aspect could be investigated through phytoplankton community characterization or by analyses of carbon

isotopes in POC. I would suggest applying the same approach to address the issue discussed in lines (326–330). In this case, $\delta^{13}C_{POC}$ could become particularly useful to distinguish POC sources and assess whether the data scattering observed in Figure 6d might reflect a case of Simpson's paradox.

**Response:** In response, we have outlined the main primary producer groups that might be contributing to the DO pool in the Danube River by incorporating information on the phytoplankton community composition from a recent Joint Danube Survey (JDS4, Liška et al., 2021). As now mentioned in the revised manuscript, diatoms were the dominant group, with co-dominance of Cyanobacteria, Chlorophyta, and Cryptophyta. These taxonomic differences likely influence spatial and temporal variations in photosynthetic activity and thus oxygen dynamics.

(326-339) "This lack of correlation may result from the stabilization of high DO values during this transition period": could the authors elaborate on this point?

**Response:** We agree that the term "stabilization" may have been misleading and have revised the discussion accordingly. In the updated version, we now refer to the gradual depletion of DO and POC concentrations during the transition period, which likely reflects a weakening of autotrophic production. Additionally, we now more clearly emphasize the potential role of increasing allochthonous inputs, that may have disrupted the observed shift in $\delta^{18}O_{DO}$ and POC.

(396-405) I would incorporate this into the "Conclusions".

**Response:** We agree and have moved this content from the Discussion section to the Conclusion.

- Technical corrections

(33) I would replace "disasters" with "stressors" or "perturbations".

**Response:** We thank the reviewer for her suggestion and have replaced "disasters" with "stressors".

(42-44) I would rephrase the second half of the sentence with the use of a semicolon.

**Response:** Again, we like to thank the reviewer for her helpful comment and we have changed the sentence accordingly.

(81) Information on data sharing on PANGAEA is sufficiently specified in "Data availability" (line 430).

**Response:** We agree with the reviewer's suggestion and have revised the sentence accordingly from this point of the manuscript.

(85) Minor issues with the legend should be fixed (e.g. brackets).

**Response:** We fixed the legend. Thank you.

(162 and 166 and 279) Please use the same number of decimal places for the same parameter (e.g. 0.4 -> 0.40).

**Response:** Following the reviewer's comment, the manuscript has been updated accordingly.

(162) This is a stylistic choice, but I would use "mmol L$^{-1}$" instead of "mmol/L".

**Response:** We appreciate the reviewer's suggestion regarding the unit formation. However, we have chosen to retain the format "mmol/L" as it is widely used and clearly understood in our field. We hope the reviewer agrees with this choice.

(170) Please ensure that the blue arrows are referenced in the legend of all plots.

**Response:** We have labeled blue (and also red) arrows in the in legend of Figures 2, 3, and 4 for clarity.

(193) Space missing between "500" and "km".

**Response:** Changed as suggested.

(210) Perhaps the sentence should be partially reworded to remove the word "created".

**Response:** We agree with the reviewer´s suggestion and replaced the word "created" with "shown" in the sentence.

(234) I suppose "enhancing" should be replaced by "enhanced".

**Response:** Changed

(263) Space missing between the references and "and".

**Response:** Changed

(303) Perhaps "and" -> "that".

**Response:** Changed

(421) "2" as subscript.

**Response:** Changed

(430) I would insert a comma (same in the supplementary materials).

**Response:** We agree with the reviewer and insert a comma instead of a period.

(521) DOI is missing.

**Response:** We apologize for missing it and added the DOI as suggested.

Comments Referee #2:

**General comments**

The manuscript "Hydrodynamic and Primary Production Effects on Seasonal DO Variability in the Danube River" by Maier et al. presents a valuable and comprehensive dataset encompassing DO, d18O_DO of DO, POC, and R/P ratios collected during five sampling campaigns along the Danube River in 2023–2024. Overall, the manuscript is well-written, and the authors provide valuable insights into the spatial and seasonal dynamics of DO and its stable isotope signatures, linking these to hydrodynamic conditions and biogeochemical drivers. However, several key aspects require clarification and further development to strengthen the manuscript's impact. 1) While the influence of the city Budapest on DO dynamics is briefly mentioned, the manuscript lacks a more comprehensive assessment of the results in context to the land use in the catchment. A discussion of how land use patterns might affect DO, d18O_DO, and R/P ratios across different regions of the river would provide meaningful context and improve the robustness of the interpretations. 2) The R/P model applied is based on a steady-state approach and includes several assumptions (e.g., constant $\alpha\_R$, absence of explicit gas exchange parameterization). These limitations should be more clearly described and critically discussed, particularly in terms of how they may affect the reliability and interpretation of the results. 3) The discussion section contains some repetitive statements that could be streamlined for clarity and conciseness.

**Response:** We thank David Piatka for his helpful and thoughtful comments and suggestions to improve our manuscript.

Regarding the major statements:

1) We have addressed land use issues in the discussion in the new version of the manuscript. Mostly, land use would exert influences due to fertilizers on the DO dynamics of the river. Here, the best proxy is $NO_3^-$. Effects and variances of this parameter has been addressed in the initial version of the manuscript and is further clarified in the new version.

2) The Danube survey was designed as a large-scale spatial study with seasonal coverage. On this scale, it was not feasible to conduct intensive diel sampling, apart from a few representative nighttime measurements (now added and discussed in the manuscript). This is a recognized limitation, and the steady-state R/P model should be viewed as a first-order approach to seasonally characterize DO dynamics in the Danube. The limited nighttime data suggest that diel variations in DO (< 0.01 mg/L) and $\delta18O_{DO}$ (<1‰) are relatively minor and were likely buffered by the large water volume. While this justifies the steady-state assumption for the main river channel, we acknowledge that in riparian zones and smaller headwater streams, diel variability is likely more enhanced. In such systems, R/P ratios would benefit from higher-resolution temporal sampling. However, we discuss these methodological limitations in the revised manuscript to clarify their potential influence on the interpretation of our results. The model we apply follows the approach introduced by Quay et al. (1995) for the Amazon Basin, another large and hydrological complex system.

3) We carefully revised the discussion section to remove redundant statements and improve clarity. Repetitive phrases and overlapping content were streamlined to enhance the flow and focus of the argumentation.

**Specific comments**

Lines 8-25: R/P ratios are missing in the abstract.

**Response:** We agree with the reviewer's comment and added the most important results of R/P ratios in the abstract.

Line 9: concentrations

**Response:** Changed

Lines 42-44: Also, input of reduced metals can play a crucial role in the DO consumption, e.g. in river systems with elevated groundwater input.

**Response:** We agree with the reviewer's suggestion and have added a generalized sentence referring to these chemical processes, such as reduced metals or redox reactions and their potential to act as DO sink.

Line 61: I suggest citing the original reference Eisenstadt et al, 2010.

**Response:** As suggested, we haves replaced the reference in the manuscript by the original reference.

Line 70-71: Incomplete sentence "approach could identify …"

**Response:** Thanks for pointing this out. We have corrected the sentence by adding "This" at the beginning and updated the manuscript accordingly.

Line 79: Can you specify in which months the sampling campaigns took place?

**Response:** The sampling campaign took place in summer (July 2023), fall (late October to early November 2023), winter (February 2024), spring (April 2024) and late summer (late August to early September 2024). This information is now included in the manuscript to improve clarity.

Line 82: Which GPS device and barometric altimeter did you use? Table 1 is missing in the Supplementary Material.

**Response:** Thank you for pointing this out. We have now specified the GPS device (Garmin eTrex HC series and Google Maps) and the Elevation App used for barometric measurements in the manuscript. A detailed list of all sampling sites, including coordinates and elevation, is

available via the PANGEA database. For this reason, it was not doubled in the Supplementary Material.

Line 84: The formatting of the date of access is not displayed correctly.

**Response:** Changed.

Figure 1: I recommend dividing the figure into two panels: A) an overview map, and B) a more detailed map. The scale in panel A is relatively small, making it difficult to read. Additionally, the legend does not explain that the thick blue line represents the Danube River. For improved clarity, it would also be helpful to indicate which tributaries were sampled.

**Response:** We have revised Figure 1 and divided it in two panels. We improved the readability of the scale of panel a) and clarified in b) that the thick blue lines represent the Danube River and explicitly labeled the samples tributaries in the caption.

Line 95: How often did you take samples from the riverbank?

**Response:** Sampling from the riverbank was only necessary when access via bridges or boats was not feasible. This was the case for 33 out of 54 sites, mainly downstream of Vienna, during the late summer 2024, fall 2023, and during winter campaigns of 2024. However, during the summer 2023 and spring 2024 campaigns, all samples were collected from bridges or boats. In all instances, we ensured that samples were collected from the flowing part of the river to maintain consistency and data quality. Further exemplary profiles across the river were able to show that even samples from side of the river sufficiently represent the main stream. We have decided not to include these logistical details in the manuscript as they do not affect the scientific outcomes.

Line 153: References to figures are usually in chronological order.

**Response:** We are aware that figure references are typically cited in chronological order. However, in this particular case, referencing Figure 6 in the "Statistical Analyses" section of the "Materials and Methods" was necessary for clarity. In all other instances, we have ensured that figures are cited in the correct chronological order.

Lines 161-162: At what time of day were the samples collected? Dissolved oxygen (DO) concentrations typically exhibit a diurnal pattern, with higher values during the day due to photosynthetic activity and lower values at night. Consequently, interpreting measured DO concentrations along a river can be misleading if sampling was conducted at different times of day. Additionally, it is unclear how long each sampling campaign lasted. Clarifying these aspects would help ensure the robustness of the data interpretation.

**Response:** We are aware that DO concentrations can exhibit diel fluctuations. However, due to the logistical constraints of our case study with large-scale river sampling, it was not feasible to sample each location at the same time of day. To assess the potential impact of diel variability, we conducted additional nighttime sampling at two sites during the late summer 2024 campaign (PANGAEA dataset). These comparative measurements showed no significant difference in DO concentrations (<0.01 mg/L) between day and night, and $\delta^{18}O_{DO}$ values increased by less than 1‰. This suggests that in large rivers such as the Danube, diel DO variability is small and falls within the range of measurement uncertainty. Overall, this timing of sample collection is an important aspect and has been taken up in the material and methods and discussion. In general, diel shifts of DO and its isotopes seem to be more important in small rivers.

Figure 2: The blue and red arrows in the figure should be labeled for clarity. While DO minima are observed in the tributaries, this pattern does not hold for the Danube River itself. To avoid confusion, it would be clearer to refer only to DO minima and maxima in the Danube River. Additionally, the figure caption should specify that the circles represent samples taken from the Danube. For the Spring 2024 sampling, some of the dashed lines indicating tributary locations are missing—this also applies to the subsequent figures and should be corrected for consistency.

Figure 3: There are additional $\delta^{18}O$ minima that are not marked with arrows. For example, in Summer 2023, there are clear minima near the confluences with the March and Váh rivers, as well as closer to the source. Similarly, in Spring 2024, minima appear around river kilometers 2300 and 2800. It is unclear why these features are not highlighted with arrows, and the rationale for selecting specific $\delta^{18}O$ minima for annotation should be clarified.

**Response:** We have labeled the red and blue arrows in Figures 2, 3, and 4 for clarity. In Figure 2, we revised the DO minima arrows to indicate features in the Danube River itself, rather than its tributaries. This approach ensures that the manuscript text also reflects this change. The figure caption has been updated to specify that circles represent samples collected from the Danube River. For Spring 2024, we added the missing dashed lines marking tributary locations to ensure consistency (Figures 2, 3 and 4). In Figure 3, we included additional $\delta^{18}O_{DO}$ minima such as those closer to the source, near the confluences with the March and Váh rivers in Summer 2023, as well as around river kilometers 2300 and 2800 in Spring 2024. We will clarify the specific $\delta^{18}O_{DO}$ minima in the manuscript and address this selection more thoroughly in the Discussion.

Lines 189-191: d18O follows the opposite trends of POC and DO.

**Response:** We agree with the reviewer's comment and clarified this aspect in the manuscript.

Line 194: The interpretation of POC concentrations in the late summer samples is somewhat unclear. Contrary to the description, relatively high POC concentrations are observed further upstream. Moreover, the peak in the middle section appears similarly high to the one observed in Summer 2023. In the lower section of the river, concentrations generally seem lower, with the exception of one prominent peak near the confluence with the Siret tributary.

**Response:** We agree with the reviewer's comment and have revised it in the manuscript.

Line 235-236: Turbulence does not increase the O2 solubility. It only accelerates the exchange of gases between the atmosphere and water.

**Response:** We agree with the reviewer and have changed it accordingly in the manuscript.

Lines 240-242: The authors are missing a clear statement of the limitations of the P/R method by Quay et al. (1995), as this is a steady-state model, and general assumptions concerning alphaR and k are met. The authors should also define what they consider to be true dominance of respiration, as this is not explicitly stated.

This point has been addressed at the beginning of the manuscript. The Danube survey was primary intended as a large-scale spatial survey with the secondary aim to address seasonality. On this scale it was not possible to perform diel sampling except for a few exemplary samples that were outlined in the text. While this aspect has only been shown with a few examples in our work we hypothesize that diel changes remain small on this large scale. However, these dynamics can become important in riparian and upstream sections. We clarified the limitations in the manuscript.

Lines 263: …(e.g., Parker et al., 2010; Wassenaar et al., 2010) and…

**Response:** Changed.

Lines 273-275: The authors should also add that such DO increases during summer by increased photosynthesis might result in low DO concentrations during night, when only respiration (by autotrophs and heterotrophs) is active.

**Response:** Although we did not conduct systematic nighttime sampling, we added a speculative statement to the discussion. Exemplary comparative measurements at two sites during the late summer 2024 campaign showed minimal day-night differences (<0.01 mg/L DO) and suggested low diel variability under our conditions.

Diel cycles have indeed be reported and are generally expected. However, whereas with e.g., CO2 it's a bit more straightforward (https://www.nature.com/articles/s41561-021-00722-3) with DO it is not that easy since it also strongly depends on location of the probe but also basic stuff like physics, the predominant community, the general conditions, discharge, morphology, pollution status etc.
See e.g., https://aslopubs.onlinelibrary.wiley.com/doi/epdf/10.4319/lo.1956.1.2.0102 and https://www.researchgate.net/publication/287738569_Dynamics_and_modelling_of_dissolved_oxygen_in_rivers https://bg.copernicus.org/articles/20/3509/2023/bg-20-3509-2023.pdf,

https://esajournals.onlinelibrary.wiley.com/doi/10.1002/ecs2.1867. In stagnant waters is relatively easy, however, in river systems (strong current, high discharge etc.) it can be a bit more complicated (https://www.sciencedirect.com/science/article/pii/S0883292723001701).

Line 281: "to a mixed signal (R/P>1)": In general, I am missing that also DO saturations are discussed, to better allow statements on photosynthesis and respiration activities.

**Response:** We have now added a sentence to clarify the role of DO saturation in supporting the interpretation of R/P ratio shifts. Specifically, we note that DO saturations increasingly fell below 100% during late summer 2024, suggesting a decline in net oxygen production and reinforcing the conclusion that respiration had begun to exceed photosynthesis.

Line 283: The authors should explain why reduced photosynthesis is causing a weakening of the correlation between DO and d18O.

**Response:** Admittedly the original statement was somewhat speculative and have now revised the text accordingly for clarity and structure. Specifically, we clarified the role of photosynthesis and respiration in two separated parts of the manuscript.

In paragraph "4.1 summer", we explain that photosynthesis contributes oxygen derived from water splitting, which does not involve isotope fractionation and thus produces a relatively stable $\delta^{18}O_{DO}$ signal (Mader et al., 2017). A strong correlation between DO concentration and $\delta^{18}O_{DO}$ can therefore reflect periods when photosynthesis is a dominant control.

In the paragraph "4.1 late summer", we explain the observed weakening of this correlation as a sign of reduced photosynthetic input and more pronounced influences by processes such as respiration or atmospheric exchange. Yet, generally respiration is carried out by diverse microbial communities and may introduce more variable isotope effects on the remaining DO pool. While more detailed studies are needed to fully quantify these effects, this shift in correlation may point to a changing balance between sources and sinks of DO. This variability may also weaken correlation between DO and $\delta^{18}O_{DO}$.

Lines 285-290: I would move this section to the conclusion part.

**Response:** We moved this section to the conclusion part.

Lines 301-302: … by atmospheric O2 exchange which serves …

**Response:** changed

Line 313: In fact, d18O is decreasing during elevated DO production.

**Response:** We thank the reviewer for this remark and apologize for this conceptual mistake. We changed it accordingly.

Lines 314-316: This is basically a repetition of what is already written in the previous chapter.

**Response:** We agree with the reviewer and removed it from this chapter.

Lines 326-327: It is not really clear what is meant by "This lack of correlation may result from the stabilization of high DO values during this transition period"

**Response:** We thank the reviewer for this helpful observation. We agree that the term "stabilization" may have been misleading and have revised the discussion accordingly. In the updated version, we now refer to the gradual depletion of DO and POC concentrations during the transition period, which likely reflects a weakening of autotrophic production. Additionally, we now more clearly emphasize the potential role of increasing allochthonous inputs, that may have disrupted the observed shift in $\delta^{18}O_{DO}$ and POC.

Lines 399-405: This section should be moved to the conclusion part. Also, pay attention to avoid repetitions.

**Response:** We agree, and also repetitions were removed.

Lines 422-428: I am missing the point that the authors are discussing the limitations of the applied R/P model. Additionally, atmospheric gas exchange (G) is a crucial (but difficult to quantify) factor for the ecosystem's health and stability, which this model cannot address. There are also other models which include G next to R and P (e.g. DOI: 10.1007/s00442-007-0744-9).

**Response:** A discussion of the limitations of the steady-state R/P model has been added to the manuscript. These include the assumption of constant atmospheric gas exchange and fixed community respiration with a generalized α_R of 0,982. In reality these parameters may vary in space and time. The model by Venkiteswaran et al. (2007) includes G but relies on diel measurements, which were not carried out in the necessary detail in our study. Few nighttime samples collected during late summer, when diel variation should be strongest, showed minimal changes in DO (< 0.01 mg/L) and $\delta^{18}O_{DO}$ (<1‰). These observations support the validity of the steady-state approach by Quay et al. (1995) for a first-order seasonal assessment.